# Drying Shrinkage and Rapid Chloride Penetration Resistance of Recycled Aggregate Concretes Using Cement Paste Dissociation Agent

**DOI:** 10.3390/ma14061478

**Published:** 2021-03-17

**Authors:** Sungchul Yang, Hyewon Lee

**Affiliations:** School of Architectural Engineering, Hongik University, 2639 Sejong-ro Jochiwon, Sejong 30016, Korea; lhw9473@g.hongik.ac.kr

**Keywords:** recycled concrete aggregate, cement paste, dissociation agent, RCA coating, mixture proportioning

## Abstract

In the present study, a recycled concrete aggregate (RCA) coating treatment using a cement paste dissociation agent (CPDA) with different mixing methods was newly incorporated in RCA concrete mixtures. First, a preliminary test program was conducted to determine the proper dosage of the CPDA solution throughout its RCA concrete test results from compressive strength, flexural strength, and elastic modulus. Then, a series of experimental tests were carried out to investigate the effect of RCA coating treatment, different mixing method such as the equivalent mortar volume (EMV) method and conventional method, and different RCA replacement ratios on durability test results of RCA concrete such as drying shrinkage values and rapid chloride penetration test (RCPT) values. The test results showed that all RCA concretes mixed with the coated RCAs were found to be workable regardless of different mix methods, with the slump and air contents of all the mixes being almost identical. All the concrete specimens, which were mixed with the coated RCAs with CPDA solution, represented lower drying shrinkage and RCPT values than those mixed without RCA coating treatment, regardless of different mix proportioning methods or RCA replacement ratios. This holds for the concrete specimens proportioned with the EMV method, regardless of different RCA replacement ratios.

## 1. Introduction

It is widely acknowledged that various types of waste materials can be transformed into recycled aggregates, powders, or additives and are used in concrete. Ceramic materials have been used as ceramic powder and ceramic aggregates in concrete [1]. This use of glass waste in concrete production and its advantages were summarized well by Zegardło et al. [2], and recent studies related to the use of rubber aggregates and chips were reported by Guettla et al. [3]. Wood chips can also be used for wood plastic composite paver blocks [4].

Recycled concrete aggregate (RCA) is one of many types of recycling aggregates that are used for concrete. RCA is known to be more porous, less dense, and more heterogeneous than natural aggregates. However, the residual mortar (RM) adhering to RCA has a negative impact on the concrete properties. Additionally, it has been reported that RM in RCA reduces the compressive strength and elastic modulus by up to 42% and 45%, respectively [5]. The data reported for Federal Highway Administration (FHWA) also revealed deceases of up to 30% and 50% in the coefficient of thermal expansion and permeability, respectively [6].

Numerous research teams have carried out experimental studies with the goal of enhancing the material properties of RCA concrete. These studies include high-quality RCA produced from precast or preserved quality concrete [7,8,9], the two-lift paving method [10], improvement of the mixing process [11,12], the use of oil-contaminated sand [13], new mixture design methods [14,15,16,17], strengthening through residual mortar (RM) coating [18,19,20,21,22], and supplementary cementitious materials.

First, high-quality RCA can be obtained from precast concrete [7], concrete sleepers, and returned concrete mixtures [8,9]. RCA acquired from such concrete can produce reliable products of consistent quality and can reduce the cost of sorting aggregates during processing.

Next, two-lift concrete paving has been successfully adopted and employed in Western Europe and the U.S. This method involves the use of low-quality RCA concrete in the lower layer of the concrete pavement. Two-lift construction using recycled concrete in the lower layer has been reported to provide the greatest impact socially and environmentally [10]. In a similar way, it was reported by Siddika et al. [23] that RCAs with other construction by-products can be used with the application of additive manufacturing, so-called, 3D printing.

The application of oil-contaminated sand in concrete was considered effective in enhancing concrete strength properties. Abousning et al. [13] reported that the presence of crude oil up to 4% could improve the properties of mortar compared to the uncontaminated samples.

After the two-stage mixing approach (TSMA) [11] was introduced, a triple mixing procedure [12] was presented for manufacturing strong and durable RCA concrete. The triple mixing process divides the mixing process into 3 steps. The first step involves coating the mixed coarse aggregate with additives and a fixed quantity of water. The second step adds cement and fine aggregates, and the last step mixes in the remaining amount of water and plasticizer. Increased density, water absorption performance, and strength properties were reported.

Improved mixing methods for RCA concrete have also been proposed by a number of other researchers [14,15,16,17]. The equivalent mortar volume method was proposed by Fathifazl et al. [14] and received significant interest. In this method, RM was considered part of the total mortar needed for concrete. Additionally, a method that processes a portion of the RM as mortar and the remainder as aggregates was presented [15]. Gupta et al. [16] proposed the equivalent coarse aggregate mass method. The major concept behind this method is processing the mortar adhered to the RCA as sand. New approaches to improving existing mixing methods have continued to be presented and consider the properties of RM of RCA after Fathifazl et al. [14], such as the identical mortar volume design method [17]. Through these new mixing methods, it was shown that the mechanical strength and drying shrinkage of RCA concrete improved to a level similar to the properties of concrete using natural aggregates.

Researchers also investigated the strengthening of RM. Various materials were used to fill pores and interfacial transition zones (ITZs) to improve RCA quality [18,19,20,21,22], including polyvinyl alcohol (PVA) solution [19], siloxane and silane polymer solutions [20], and pozzolanic material slurry [21]. Overall, the surface treatment materials mentioned above helped reduce water absorption in RCA. Recently, the bio-deposition method was introduced as a way to enhance RCA quality [22]. This method used bacteria to produce calcium carbonate on the cell surfaces near the pores of ITZs (when appropriate calcium sources existed). Water absorption and compressive strength were improved using this method.

Drying shrinkage is a very important property of cementitious composites influencing their durability. Several models to predict the concrete drying shrinkage using influential factors have been empirically proposed in terms of slump, air content, fine aggregate, cement content, compressive strength, relative humidity, and volume-to-surface ratio, etc. [24,25]. In addition, it was reported that RCA treatment by PVA solution [19] affects concrete drying shrinkage. It has been widely accepted that the use of more RCA in the conventional concrete mix leads to an increase in drying shrinkage [5,6,8,9,11,17]. Some researchers summarized that drying shrinkage of the RCA concrete exhibited a 6–111% increase in 11 studies [5] while a 20–50% increase for coarse RCA and 70–100% increase for coarse and fine RCA in concrete pavement research areas [6] compared to that of the natural aggregate concrete.

Chloride penetration resistance, which is a measure of concrete durability, is important in reinforced concrete structures as well as plain concrete structures that do not use rebars. The presence of chloride ions was found to affect the spalling of concrete pavement below the freezing temperature of the pore solution and increased the saturation state of rebar corrosion [26]. The Rapid Chloride Penetration Test (RCPT), which is another such test, provides a diffusion-related conductivity measurement and index, where a higher RCPT value indicates greater concrete diffusion [26]. Factors that affect the RCPT include the compressive strength [26]; mineral admixtures such as fly ash [27,28,29]; aggregates [8,30,31,32,33]; curing conditions such as autoclaving, steam curing, and normal curing [34]; pore size [26,27]; RCA replacement ratio [35,36,37]; specimen temperature [38,39,40]; regional environment conditions [26]; test conditions [40]; multiple-stage mixing approaches [41,42]; and sodium silicate and silica fume coating of the RCA aggregate [41,42]. Especially, the RCA coating was observed to fill the cracks and pores using the ITZ of the concrete sample through microstructural analysis [41,42].

The above studies highlight the importance of an idealized mixing process, a mix proportioning method, ITZ strengthening, and a coating method to provide enhanced mechanical properties for RCA concretes. However, most previous studies have been limited to investigating the mechanical strength properties of RCA concretes. Readers may notice that, so far, no research determining the effect of RCA coating treatment by cement type solutions has been conducted on durability properties such as drying shrinkage and chloride ion penetration resistance. Thus, in the present study, an RCA coating treatment using a cement paste dissociation agent (CPDA) with different mixing methods was newly incorporated in RCA concrete mixtures. First, a preliminary test program was conducted to determine the proper dosage of CPDA solution using its RCA concrete test results on compressive strength, flexural strength, and elastic modulus. Then, a series of experimental tests were carried out to investigate the effect of RCA coating treatment, different mixing methods such as the equivalent mortar volume (EMV) method and conventional method, and different RCA replacement ratios on the durability test results of RCA concrete such as drying shrinkage values and RCPT values [43]. Therefore, the results of this study provide guidance that can be used to assess the beneficial increment in durability properties by adopting RCA coating treatment with the optimized mix proportioning method.

## 2. Experimental Program

### 2.1. RCA Production

This experimental study used RCAs produced from two different sources in South Korea. The RA aggregate was crushed with a maximum size of 25 mm from old runway concrete pavement at an air base reconstruction site. The RP was obtained with a maximum size of 20 mm by crushing the precast concrete (PC) culverts (see Figure 1). It should be noted that the first letter R from RA and RP refers to recycled coarse aggregate and the second letter denotes aggregate types used in this study where A is an arbitrary symbol while P was named since RP was manufactured from precast concretes.

The PC culverts had a compressive strength of 35 MPa with a maximum aggregate of 20 mm and a water–cement ratio of 35% or lower according to the Korean Construction Specification [44] and were manufactured by steam curing.

### 2.2. Aggregate Properties

Table 1 shows the specific gravity, absorption rate, and residual mortar content (RMC) of the RCA and the specific gravity and absorption rate of the natural coarse aggregate and fine aggregate. The specific gravity and absorption ratio of the RA were 2.54 and 4.81%, respectively. The specific gravity and absorption ratio of the RP were 2.60 and 2.62%, respectively.

Crushed granite (which was the same source of aggregate previously used in PC culvers) was used as the natural coarse aggregate, and its specific gravity and absorption rate were 2.69 and 0.54%, respectively. Natural river sand was used as the fine aggregate. RA and fine aggregate 1 were used in the first mix series, while RP and fine aggregate 2 were used in the second mix series. Figure 2 shows the particle size distribution for the RP, natural coarse aggregate (NCA), and fine aggregate 2, satisfying Korean standards.

In order to obtain the RMC value for *RP*, the heat treatment method proposed by Juan and Gutierrez [45] was used. After heating the prepared RCA samples for 2 h at 500 °C in an electric furnace, the heated samples were removed from the furnace and immersed in ice water to subject the samples to thermal shock and to separate the mortar. The RMC value was calculated by substituting the measured values in Equation (1). The RMC value was 20.0%.
RMC = (W_RCA_ − W_OVA_)/W_RCA_ × 100(1)
where W_RCA_ is the weight of the RCA dried in an oven dryer after first collecting the samples and W_OVA_ is the weight of the original virgin aggregate (OVA) dried in an oven dryer after removing residual mortar.

### 2.3. Cement Paste Dissociation Agent

The cement paste dissociation agent (CPDA) used in this study was a product composed of SiO_2_, CaO, NaCl, NaNO_3_, Na_2_SO_4_, and K_2_CO_3_. The chemical equations below explain the additional hydration reaction of calcium hydroxide from the cement and the CPDA [46]. Free calcium oxide of cement forms calcium hydroxide when mixed with water. Then, calcium hydroxide takes part in the reactions with the second and forth components to the left in the following equations, which are the main components of the CPDA to further form inorganic crystallization or ettringite around old ITZs or new ITZs.
3Ca(OH)_2_ + 6NaCl + 30H_2_O + (3CaOAl_2_O_3_) ⇒ 3(CaOAl_2_O_3_CaCl_2_10H_2_O) + 6NaOH(2)
3Ca(OH)_2_ + 6NaNO_3_ + 32H_2_O + Ca_3_(AlO_3_)_2_ ⇒ 3Ca_3_(AlO_3_)_2_3Ca(NO_3_)_2_32H_2_O + 6NaOH(3)
3Ca(OH)_2_ + 3Na_2_SO_4_ + 31H_2_O + Ca_3_(AlO_3_)_2_ ⇒ 3(CaOAl_2_O_3_)_2_CaSO_4_31H_2_O + 6NaOH(4)

As a percentage of cement, 1.25% CPDA powder was recommended for conventional cement concretes by the CPDA manufactured company [46]. However, in this study, to coat the weak parts of old ITZs in the RCA concrete, 2.5% CPDA powder as a cement percentage, which is two times higher than the recommended dosages, was selected. Thus, for a preliminary test program (first mix series in Table 2), 0, 2.5, and 5% CPDA powder as a percentage of cement amounts were added in RCA concrete mixes. The test results from Section 4.1 showed that the addition of 2.5% CPDA as chemical admixtures in the RCA concrete mixes helps improve compressive strength, flexural strength, and elastic modulus properties. Thus, in the main test program (second mix series in Table 2), 300 g of CPDA was diluted in 1 L of water instead of powder but with the same dosage and sprayed on the RCA surface as shown in Figure 3. The coated RCAs were poured in a pan and covered with plastic for curing for 2 days. Then, they were used in the concrete mixture.

## 3. Experimental Tests

### 3.1. Mix Design

In this experiment, type I Portland cement was used with a specific gravity of 3.15 and specific surface area of 3380 cm^2^/g. The chemical admixture used in this experiment was an air entraining and water reducing agent solution. All of the aggregates were prepared in a saturated surface-dry condition.

Two series of mixes were prepared and tabulated in Table 2. The first series of mixes were designed in order to investigate the optimal amounts of cement paste dissociation agent (CPDA) powder. In the nomenclature from the first mix series, the numbers denote CPDA contents as a percentage of cement amounts.

The second series of mixes were then designed to find whether coating the RCAs with CPDA solution is an effective way to improve the durability properties such as drying shrinkage and chloride penetration resistance as well as mechanical strength properties of recycled aggregate concrete (RAC). This mix series involved nine mixtures. The conventional American Concrete Institute (ACI) mixing method and the equivalent mortar volume (EMV) mixing method were used. In the nomenclature, first, C and E in mix identification denote conventional and equivalent mix design, respectively. Next, R and N denote natural coarse aggregate and RCA, respectively. Thirdly, A refers to type A aggregate while P refers to aggregate manufactured from precast concretes. Fourthly, -a and -b refer to the 25% and 50% RCA replacement ratios per total coarse aggregate, respectively. Lastly, S used in the mixture nomenclature refers to the use of spray coating of the recycled concrete aggregate (RCA) surface in the mixture.

A pan mixer that could perform 60 L volume mixing was available in a research laboratory from Hongik University in Sejong, Korea, which is where the experimentation was conducted. First, the coarse aggregate and fine aggregate were poured in the mixer and mixed together for around 30 s. Then, cement was added and mixed for 30 s. Afterwards, the admixture was diluted sufficiently in water and added to the mixture for additional mixing for around 2 min.

### 3.2. Specimen Preparation

The specimens for concrete compressive strength measurement were prepared using a 100 mm × 200 mm plastic mold following a predetermined method [47]. The fabricated specimens were removed from the mold after 24 h and moist cured at approximately 20 ± 2 °C until the compressive strength testing. Three specimens were tested for each mixture, and their average values were calculated.

Drying shrinkage experiments were performed using a dial gauge, as suggested by KS (Korean Standards) F 2424 [48], which is equivalent to ASTM (American Society for Testing and Materials) C 157-08 [49]. Two rectangular specimens with 100 × 100 × 400 mm were prepared for each mixture. The drying shrinkage strain was measured by the dial gauge. The specimens were kept inside an environmental chamber, which was controlled at 20 °C and 60% Relative Humidity (RH). Figure 4 shows a schematic diagram and experimental specimens with the dial gauges installed.

In order to test the chloride ion penetration resistance, three cylindrical specimens with a 100 mm diameter and 200 mm height were prepared, followed by fabrication into disc specimens of 50 mm thickness in accordance with ASTM C 1202 [50]. The prepared disc specimens were placed in a vacuum state maintained for 3 h; then, the specimens were immersed in distilled water for 18 h to fulfill the saturation state pretreatment. For the 100 mm diameter and 50 mm thickness specimens, the cathode was filled with 3% NaCl aqueous solution and the anode was filled with 0.3 M NaOH aqueous solution followed by the application of 60 V for 6 h. Figure 5 shows a schematic diagram for the Rapid Chloride Penetration Test (RCPT) setup. After the initial current measurement, the passing current value was measured in 30-min intervals and the total passed charge was calculated using Equation (5).
*Q* = 900(*I*_0_ + 2*I*_30_ + 2*I*_60_ + ⋯ + 2*I*_300_ + 2*I*_330_ + *I*_360_)(5)
where *Q* refers to the total passed charge and *I*_*n*_ refers to the current at *n* minutes.

## 4. Experimental Test Results

### 4.1. Preliminary Test Results from First Mix Series

Figure 6 shows average mechanical concrete strength properties such as compressive strength, flexural strength, and elastic modulus with error bars of the first mix series. In Figure 6a, the compressive strengths increased by 32% and 24% at 7 and 28 days, respectively, in CRA-2.5 mix compared to CRA-0 mix (control specimen). On the other hand, the compressive strengths in CRA-5 mix decreased by 27% and 24% at 7 and 28 days, respectively. The flexural strength test results showed a similar pattern to the compressive strength test results. The flexural strengths increased by 56% and 18% at 7 and 28 days, respectively, in CRA-2.5 mix compared to the control mix, whereas the flexural strengths in CRA-5 mix decreased by 21% and 3% at 7 and 28 days, respectively. Figure 6c shows the elastic modulus of the concrete test results at 28 days. In a similar manner to the compressive and flexural strength test results, elastic modulus increased by 3% in CRA-2.5 mix but decreased by 9% in CRA-5 mix compared to the control specimen. Thus, it appears that the addition of 2.5% of cement paste dissociation agent (CPDA) in the recycled concrete aggregate (RCA) concrete mixes helps improve the mechanical strength properties.

### 4.2. Test Results from Second Mix Series

#### 4.2.1. Fresh and Hardened Properties

It is commonly accepted that concrete durability is thoroughly related to fresh and hardened properties. Table 3 presents the measured fresh and hardened concrete properties from the second mix series. Generally, concrete, which is prepared with workable slump and proper air content and is compacted with high density, is expected to be durable with reasonable compressive strength. It was observed from Table 3 that all RCA concrete mixes proportioned with the conventional mix design method and the equivalent mortar volume (EMV) design method were found to be workable, with their slump ranging between 150 and 160 mm. The air contents of all the mixes were almost identical, ranging from 3.9% to 4.2%. Thus, it can be inferred that slump or air content of RCA concretes from the second mix series does not affect the mechanical strength or durability of RCA concretes.

Table 3 summarizes density values at the fresh state and hardened state. While there were some deviations depending on the degree of aggregate coating, RCA substitution ratio, and fresh or hardened state, it seems that the densities of the ERP mixes are 1–2% lower values compared to that of the CNC and CRP mixes.

The compressive strength values at 28 days are represented in Table 3. Excluding ERP-a and ERP-bS, the compressive strength values were similar, at around 30.1–32.2 MPa. It was predicted [14,15] that the concrete that uses the EMV mixing method would exhibit higher strength than concrete that uses the conventional volume mixing method. However, in the case of the RCA used in this mixture, its quality was excellent, with a specific gravity of 2.60 and absorption rate of 2.62%, resulting in favorable strength even when the conventional ACI volume mixing method was used.

Figure 7 shows the average Young’s modulus of the concrete samples at 28 days. All the elastic modulus values ranged from 26.3–27.7 GPa within 5% difference, regardless of different mixing methods or use of coating treatment method.

#### 4.2.2. Drying Shrinkage

The drying shrinkage test results are shown in Figure 8. Drying shrinkage tests were conducted for all the specimens initially for 111 days, as shown in Figure 8a. It must be noted that the initial wet curing period of 8 days instead of the standard wet curing period of 7 days was employed for the control specimen (CNC) and CRP specimens by malfunction of the environmental chamber. Therefore, it can be seen in Figure 8a that the ERP-a specimen produced higher drying shrinkage test values at early ages compared to the CRP specimens. However, previous researches [5,9,15] revealed that the EMV mixing method yielded a drying shrinkage property of the RCA concrete lower than that of the RCA concrete mixed with the conventional mixing method. Therefore, to verify that this comparatively higher shrinkage value of the ERP at early stages becomes a gradually lower value than that of the CRP specimens, drying shrinkage strains of the CNC, CRP-a, CRP-b, and ERP-a specimens were further measured for 648 days, as shown in Figure 8b. The test results are subsequently discussed in terms of three influencing factors: (1) mix proportioning method, (2) with and without RCA coating treatment, and (3) RCA replacement ratio.

##### Dependence of Mix Proportioning Method

The effect of different mix proportioning methods on drying shrinkage was studied, and their average test results are plotted with error bars in Figure 9. First, as shown in Figure 9a, the drying shrinkage values of CRP-a and ERP-a specimens were almost identical at 111 days, as mentioned before, but the drying shrinkage of the ERP-a specimen decreased by 4.6% at 648 days compared to that of CRP-a specimen. Likewise, in Figure 9b–d, the drying shrinkage values of the ERP-aS, ERP-b, and ERP-bS specimens decreased by 5.1%, 9.6%, and 0.1% compared to those of the CRP-aS, CRP-b, and CRP-bS specimens at 111 days, respectively. Hence, a lower drying shrinkage of RCA concrete can be achieved using the EMV mix proportion method compared to the RCA concrete proportioned by the conventional mix design method. This was confirmed by Fathifazl et al. [5] that drying shrinkage is proportional to the total volume of the mortar and that the conventional RCA concrete mix contains a higher volume of mortar due to residual mortar.

##### Dependence of Coating Treatment

The effect of RCA coating on drying shrinkage was studied, and their average test results are plotted in Figure 10. Figure 10a–d show that the drying shrinkage values of CRP-aS, CRP-bS, ERP-aS, and ERP-bS mixes at 111 days decreased by 1%, 11%, 6%, and 2%, respectively, compared to those of CRP-a, CRP-b, ERP-a, and ERP-b. Thus, all the concrete specimens, which were mixed with coated RCA, represented better drying shrinkage performance than those mixed without RCA coating treatment, regardless of different mix proportioning methods and RCA replacement ratios.

##### Dependence of RCA Replacement Ratio

The test results were compared to investigate the effect of the RCA replacement ratios on the drying shrinkage of RCA concretes. In Figure 11a, compared to the drying shrinkage value of the CRP-a concrete specimen, which is made with 25% RCA replacement ratio, that of the CRP-b concrete specimen with 50% RCA replacement ratio was increased marginally by 2% at 111 days. However, reversed test results were observed in Figure 11b–d. In Figure 11b, CRP-bS specimens resulted in an 8.5% drop in concrete drying shrinkage compared to CRP-aS specimens. Although CRP-bS specimens were mixed with a 50% RCA replacement ratio using the conventional mix proportioning method, double the amount of coated RCAs led to a drop in drying shrinkage value compared to CRP-aS specimens. It is generally accepted that more RCA replacement in concrete mix leads to inferior drying shrinkage performance. However, the little increase or reversed result of drying shrinkage in concrete made with double the amount of RCA replacement may be attributed to the good-quality RCA adopted in this study. In Figure 11c, an 8% drop in concrete drying shrinkage of the ERP-b specimen was obtained with a 50% RCA replacement ratio compared to that of the ERP-a specimen with a 25% RCA replacement ratio. It should be remembered that the drying shrinkage values of the EMV mixes are expected to be similar regardless of different RCA replacement ratios incorporated in the EMV mix design, as the total mortar is the same. In a similar manner, about a 4% decrease in drying shrinkage for the ERP-bS specimen was observed in Figure 11d compared to the ERP-aS specimen.

#### 4.2.3. Chloride Ion Penetration Resistance

Figure 12 shows the chloride ion penetration resistance experiment results for the concrete specimens. In the same manner as in Section 4.2.2, the test results are subsequently discussed in terms of three influencing factors: (1) mix proportioning method, (2) with and without RCA coating treatment, and (3) RCA replacement ratio.

ASTM C 1202 [50] recommends the total charge passed between 2000 to 4000 C to be a moderate condition for concrete specimens. From this recommendation, except for CNC, CRP-a, and CRP-b, all other mixes in Figure 12 represented good resistance against the chloride ion penetration resistance. Thus, it may be said that the ERP mixes have unparalleled resistance regardless of RCA coating treatment, but combined with the RCA coating treatment, it produced the finest performance against chloride ion penetration.

##### Mix Proportioning Method

The effect of different mix proportioning methods on chloride ion penetration resistance was studied and their Rapid Chloride Penetration Test (RCPT) results are plotted in Figure 13. In Figure 13a–d, the RCPT values of the ERP-a, ERP-aS, ERP-b, and ERP-bS specimens decreased remarkably by 30%, 19%, 40%, and 28%, compared to those of the CRP-a, CRP-aS, CRP-b, and CRP-bS specimens, respectively. In fact, the RCPT values for RCA concrete specimens proportioned by the EMV method were substantially lower than those of the specimens made of a mixture proportioned by the conventional method. Thus, it may be inferred that the RCPT values are related to the total mortar volume of the RCA concrete mix since the total mortar volume in the ERP mixes is reduced due to the residual mortar attached to the RCAs.

##### Coating Treatment

The effect of RCA coating on the RCPT results was studied, and their average test results are plotted in Figure 14. Figure 14a–d show that the RCPT values of CRP-aS, CRP-bS, ERP-aS, and ERP-bS mixes decreased by 21%, 37%, 10%, and 24%, respectively, compared to those of CRP-a, CRP-b, ERP-a, and ERP-b. Thus, all the concrete specimens, which were mixed with the coated RCAs, represented reduced RCPT values compared to those mixed without RCA coating treatment, regardless of different mix proportioning methods or RCA replacement ratios. In fact, implementing the RCA coasting treatment using the CPDA solution before concrete mixing resulted in better performance for drying shrinkage as well as chloride ion penetration resistance.

##### RCA Replacement Ratio

The test results were compared to determine the effect of RCA replacement ratios on the RCPT values of RCA concretes, as shown in Figure 15. In Figure 15a, compared to the RCPT value of the CRP-a concrete specimen, which is made with 25% RCA replacement ratio, that of the CRP-b concrete specimen with 50% RCA replacement ratio increased by 11%. It is also widely accepted that more RCA replacement in concrete mix leads to inferior chloride ion penetration resistance. However, contrary test results were observed in Figure 15b–d. In Figure 15b, the CRP-bS specimens resulted in a 12% drop in the RCPT value compared to CRP-aS specimens. Although CRP-bS specimens were mixed with 50% RCA replacement ratio, using the conventional mix proportioning method, double the amount of coated RCAs led to a drop in the RCPT value compared to CRP-aS specimens. In Figure 15c, a 6% drop in the RCPT value of the ERP-b specimen was obtained with 50% RCA replacement ratio compared to that of the ERP-a specimen with 25% RCA replacement ratio. As mentioned before, those two test values were supposed to be similar, regardless of different RAC replacement ratios, since their total mortar is the same. Likewise, about a 20% drop in the RCPT value for the ERP-bS specimen was observed in Figure 15d compared to ERP-aS specimen. Similar to the test results from Section 4.2.2 Drying Shrinkage, the reversed result of the RCPTs in concrete made with double the amount of RCA replacement except for CRP-a versus CRP-b may be attributed to the good-quality RCA adopted in this study. Further study on this effect is necessary.

#### 4.2.4. Micro Structural Analysis of RCA

A scanning electronic (SEM) microscope was used to observe the microstructural characteristics of RCA concretes. In this study, CRP-b and CRP-bS samples only were tested to check whether the CPDA coating effect was seen in the image analysis. In Figure 16b, the darker area is the original virgin aggregate, while the lighter part in Figure 16b is old mortar. The strip area between aggregate and old mortar is the interfacial transition zone (ITZ). It can be seen that the old ITZ in CRP-b (Figure 16a) is different from the old ITZ in CRP-bS (Figure 16b). The old ITZ width in Figure 16a ranges between 22 to 47 μm, while the old ITZ width in Figure 16b ranges between 7 to 11 μm. This implies that the coating effect of the CPDA can modify the microstructure of ITZs. Further research is needed to explore image analysis related to this topic.

## 5. Conclusions

In this study, recycled concrete aggregate (RCA) manufactured from precast concrete (PC) culvert was used to carry out durability experiments on concrete fabricated using the conventional ACI volume mixing method and the equivalent mortar volume (EMV) mixing method. Moreover, a cement paste dissociation agent (CPDA) was used to coat the RCA surface followed by curing to investigate the drying shrinkage and the chloride ion penetration resistances of the concrete coated at the RCA surface with CPDA compared with concrete that did not. The following conclusions were obtained from the experimental results.

From the preliminary test, it appears that the addition of 2.5% of CPDA in the RCA concrete mixes helps improve the mechanical strength properties such as compressive strength, flexural strength, and elastic modulus. The test results showed that about 24%, 18%, and 3% increases were observed for compressive strength, flexural strength, and elastic modulus, respectively, at 28 days.All RCA concretes mixed with coated RCA were found to be workable regardless of different mix methods, with the slump and air contents of all the mixes being almost identical. Additionally, all the concrete specimens that were mixed with the coated RCAs with CPDA solution had lower drying shrinkage values and RCPT (Rapid Chloride Penetration Test) values than those mixed without RCA coating treatment, regardless of different mix proportioning methods or RCA replacement ratios. The variation in compressive strength and density of the RCA concrete did not affect drying shrinkage or RCPT values.This holds for the concrete specimens proportioned with the EMV method, regardless of different RCA replacement ratios. Especially, it may be inferred that the RCPT values are related to total mortar volume of RCA concrete mix since total mortar volume in the ERP mixes is reduced due to the residual mortar attached to the RCAs.For the conventional RCA mixes, in comparison to test results from the 25% RCA replacement ratio, the little increase or reversed test result of drying shrinkage values and RCPT values in concrete made with 50% RCA replacement ratio may be attributed to the good-quality RCA adopted in this study. Further study on this effect is necessary.

## Figures and Tables

**Figure 1 materials-14-01478-f001:**
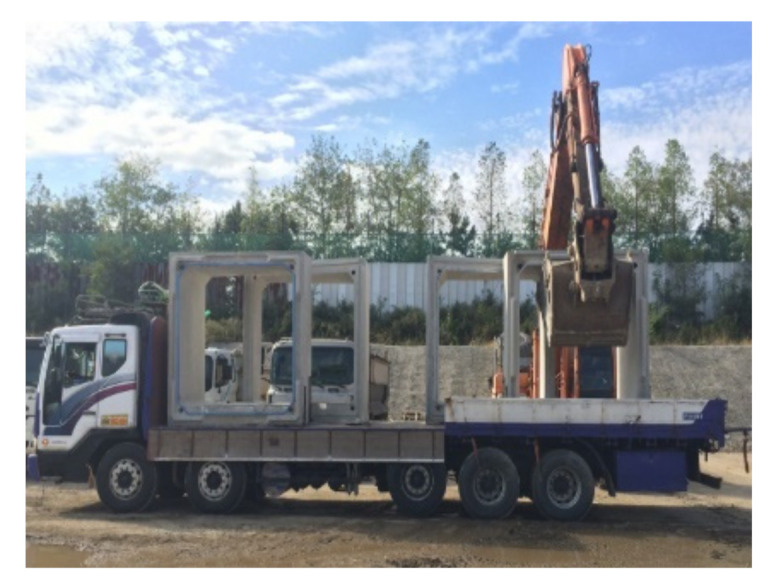
Recycled concrete aggregate (RCA) sources.

**Figure 2 materials-14-01478-f002:**
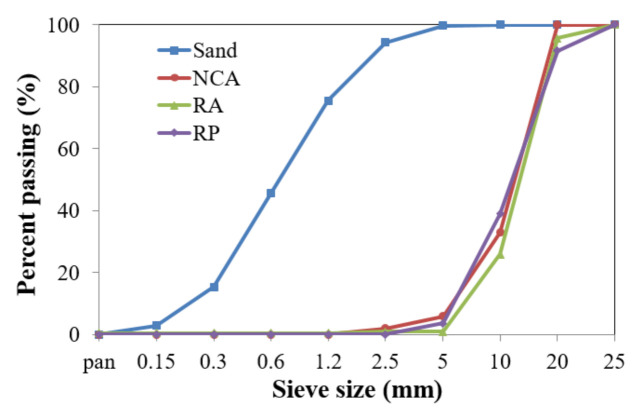
Aggregate gradation for RP, natural coarse aggregate (NCA), and fine aggregate 2.

**Figure 3 materials-14-01478-f003:**
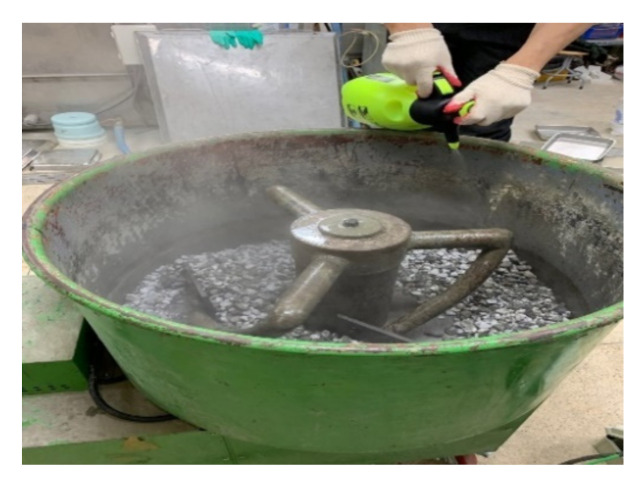
Spray coating RCAs.

**Figure 4 materials-14-01478-f004:**
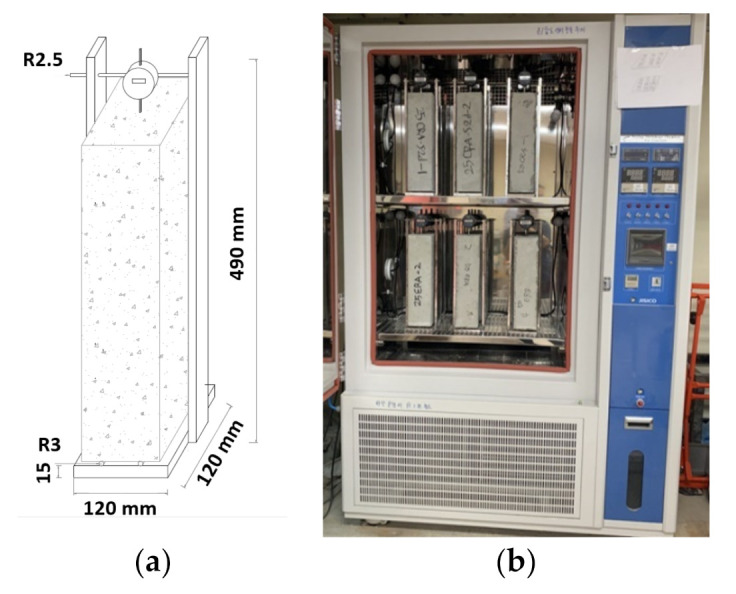
Experimental specimens with a dial-gauge installed: (**a**) schematic diagram and (**b**) picture.

**Figure 5 materials-14-01478-f005:**
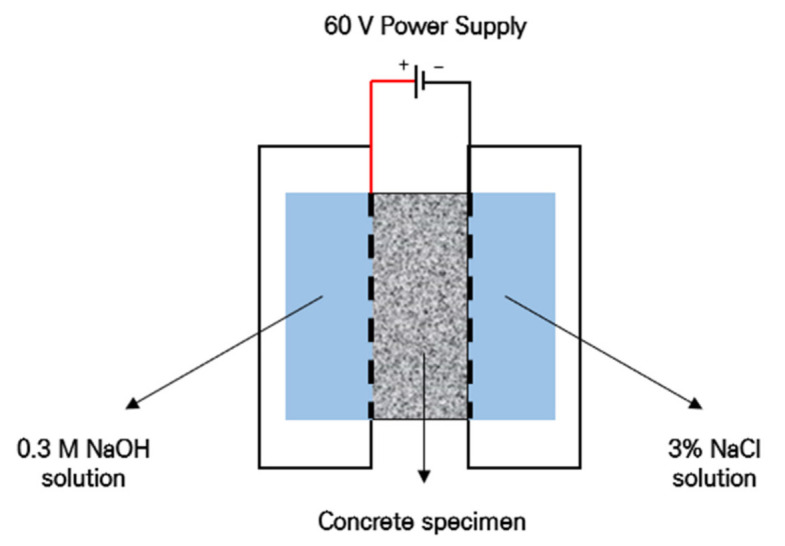
Chloride ion penetration resistance test setup.

**Figure 6 materials-14-01478-f006:**
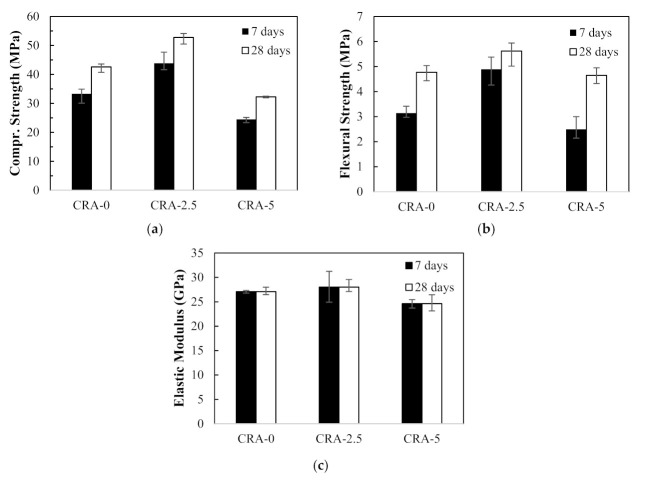
Mechanical strength results of first mix series: (**a**) compressive strength, (**b**) flexural strength, and (**c**) elastic modulus.

**Figure 7 materials-14-01478-f007:**
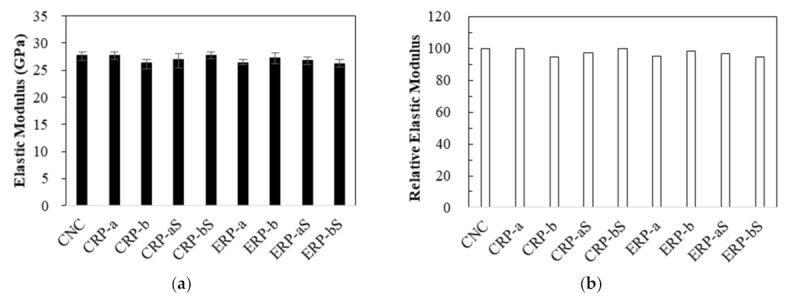
Elastic modulus and relative values: (**a**) elastic modulus and (**b**) relative elastic modulus.

**Figure 8 materials-14-01478-f008:**
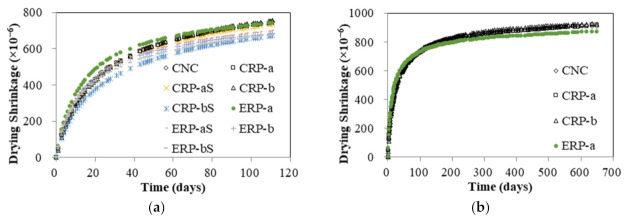
Drying shrinkage test results: (**a**) all specimens tested for 111 days and (**b**) some specimens tested for 648 days.

**Figure 9 materials-14-01478-f009:**
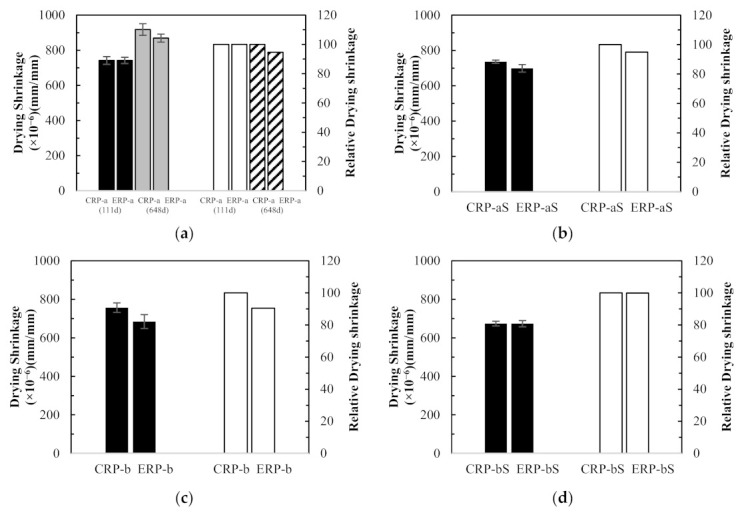
Concrete drying shrinkage results and relative values between different mix design method: (**a**) CRP-a versus ERP-a, (**b**) CRP-aS versus ERP-aS, (**c**) CRP-b versus ERP-b, and (**d**) CRP-bS versus ERP-bS.

**Figure 10 materials-14-01478-f010:**
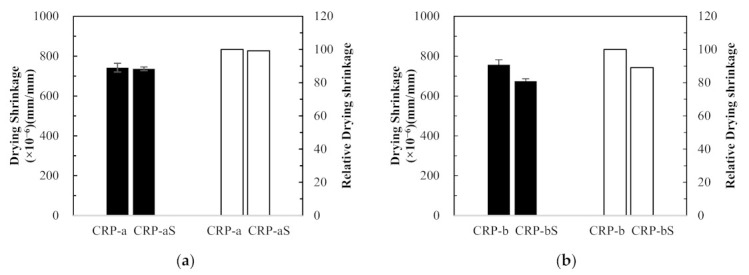
Concrete drying shrinkage results and relative values with and without recycled concrete aggregate coating treatment: (**a**) CRP-a versus CRP-aS, (**b**) CRP-b versus CRP-bS, (**c**) ERP-a versus ERP-aS, and (**d**) ERP-b versus ERP-bS.

**Figure 11 materials-14-01478-f011:**
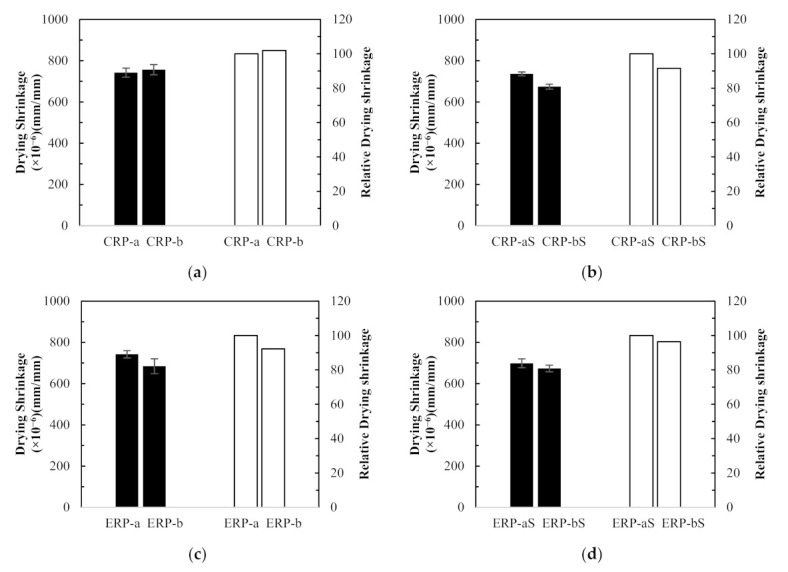
Concrete drying shrinkage results and relative values between different recycled concrete aggregate replacement ratios: (**a**) CRP-a versus CRP-b, (**b**) CRP-aS versus CRP-bS, (**c**) ERP-a versus ERP-b, and (**d**) ERP-aS versus ERP-bS.

**Figure 12 materials-14-01478-f012:**
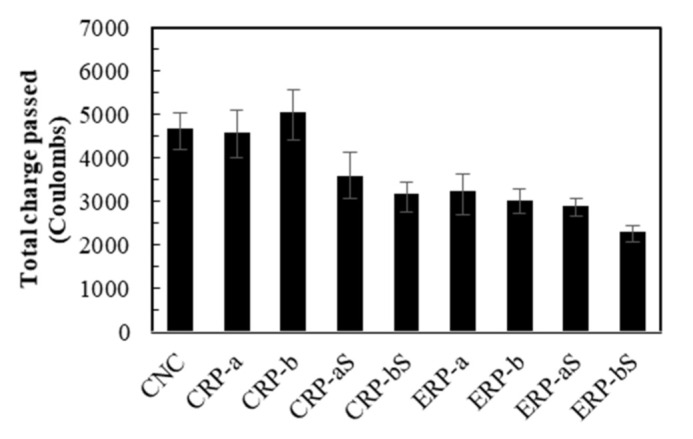
Chloride ion penetration resistance experiment results.

**Figure 13 materials-14-01478-f013:**
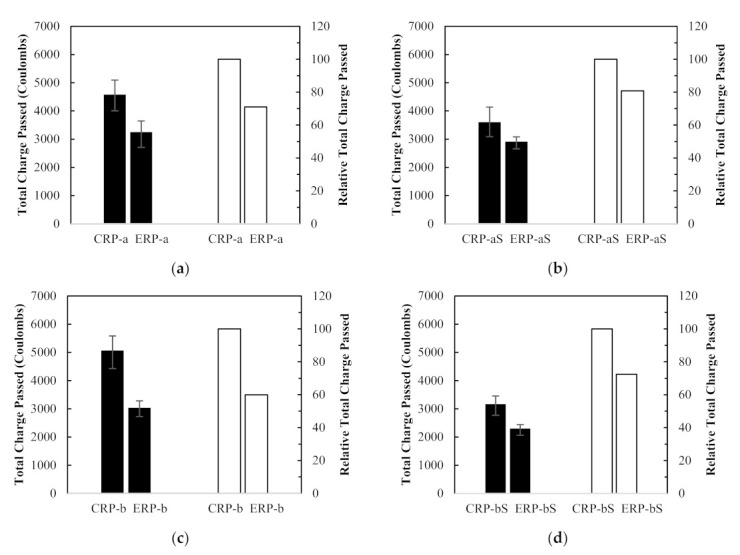
Rapid Chloride Penetration Test results and relative values between different mix design method: (**a**) CRP-a, (**b**) CRP-aS, (**c**) CRP-b, and (**d**) CRP-bS.

**Figure 14 materials-14-01478-f014:**
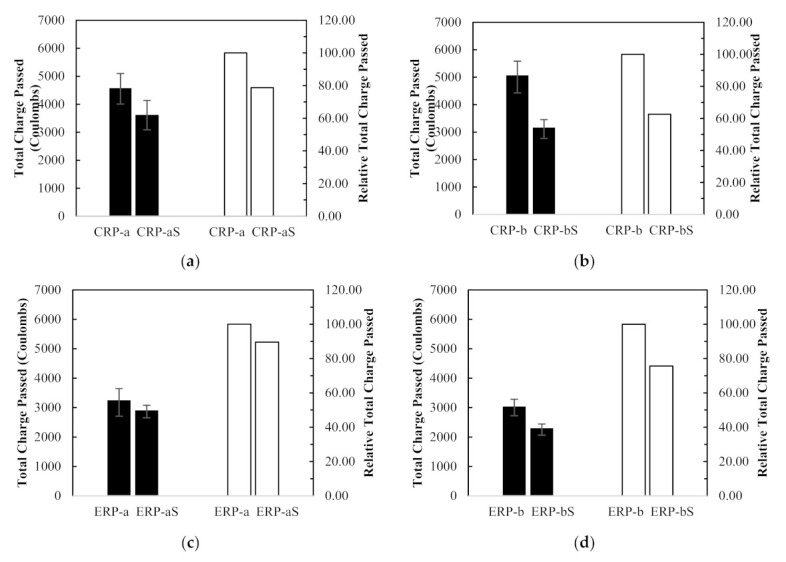
Rapid Chloride Penetration Test results and relative values between different mix design method with and without recycled concrete aggregate coating: (**a**) CRP-a, (**b**) CRP-b, (**c**) ERP-a, and (**d**) ERP-b.

**Figure 15 materials-14-01478-f015:**
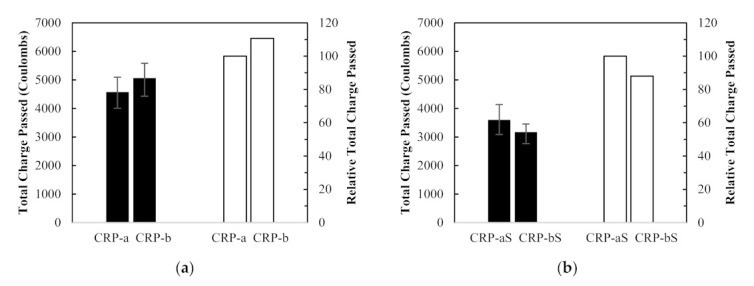
Rapid Chloride Penetration Test results and relative values between different recycled concrete aggregate replacement ratios: (**a**) CRP-a, (**b**) CRP-b, (**c**) ERP-a, and (**d**) ERP-b.

**Figure 16 materials-14-01478-f016:**
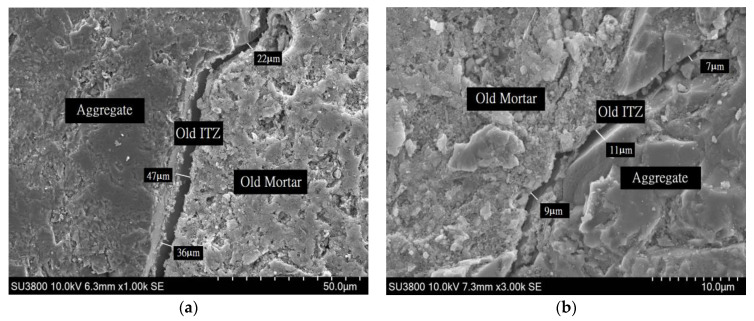
Scanning electronic microscope in the interfacial transition zone (ITZ) of different RCAs: (**a**) CRP-b and (**b**) CRP-bs.

**Table 1 materials-14-01478-t001:** Basic aggregate properties.

Test Items	RA	RP	NCA	Fine Aggregate 1	Fine Aggregate 2
Specific gravity	2.54	2.60	2.69	2.58	2.60
Absorption rate (%)	4.81	2.62	0.54	0.52	0.95
RMC	-	20.0	-	-	-

**Table 2 materials-14-01478-t002:** Concrete mixture designs and material quantities.

Test Series	Mix	W/C	S/a	RCA wt %	Mix Proportions (kg/m^3^)							
					W	C	S	F/A	NCA	RCA	CPDA	Admixture
1	CRA-0	0.37	38.9	100	138	370	695	-	-	1093	0	-
	CRA-2.5	0.37	38.9	100	138	370	695	-	-	1093	9.25	-
	CRA-5	0.37	38.9	100	138	370	695	-	-	1093	18.5	-
2	CNC	0.36	39.1	0	158	396	675	44.0	1051	0	-	2.56
	CRP-a	0.36	40.4	25	158	396	694	44.0	767	256	-	2.49
	CRP-b	0.36	41.7	50	158	396	712	44.0	499	498	-	2.49
	CRP-aS	0.36	40.4	25	158	396	694	44.0	767	256	-	2.49
	CRP-bS	0.36	41.7	50	158	396	712	44.0	499	498	-	2.49
	ERP-a	0.36	37.0	25	152	380	648	42.3	830	276	-	2.60
	ERP-b	0.36	34.6	50	145	363	619	40.3	584	583	-	2.61
	ERP-aS	0.36	37.0	25	152	380	648	42.3	830	276	-	2.74
	ERP-bS	0.36	34.6	50	145	363	619	40.3	584	583	-	2.62

**Table 3 materials-14-01478-t003:** Fresh and hardened properties of the mixes.

Mix iD	Fresh Property			Hardened Property		
	Slump (mm)	Air Content (%)	Density (kg/m^3^)	Density (kg/m^3^)	Compressive Strength (MPa)	Coefficient of Variation (%)
CNC	155	3.9	2414	2334	32.2	3.00
CRP-a	150	4.0	2333	2428	31.6	2.60
CRP-b	155	4.1	2360	2332	30.8	3.67
CRP-aS	155	4.0	2364	2301	30.3	4.84
CRP-bS	150	4.2	2389	2389	31.6	2.25
ERP-a	160	4.1	2294	2267	27.7	2.00
ERP-b	150	4.2	2336	2336	30.5	3.68
ERP-aS	155	4.1	2354	2306	30.1	2.53
ERP-bS	150	4.3	2314	2249	24.8	2.48

## Data Availability

The data presented in this study are available on request from the corresponding author.

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
