# Peer review of "Drying Shrinkage and Rapid Chloride Penetration Resistance of Recycled Aggregate Concretes Using Cement Paste Dissociation Agent"

_materials, 2021, doi:10.3390/ma14061478_

Round 1

Reviewer 1 Report

Please, change the title of the paper. The term RCA used is not explained in the title. The readers can not know the meaning. In the same way, add the explanation of acronyms in the abstract.

On the other hand, the paper is clearly exposed, and well structured. The authors provide a wide study about an RCA coating treatment by a cement paste dissociation agent (CPDA) with different mixing methods to incorporate in RCA concrete mixtures. The main question is why this cement paste dissociation agent used? Can the authors explain the composition? Preliminary studies should be more detailed to know how the authors consider that formulation. 

The nomenclature used by the authors is difficult to understand. A list of acronyms should be provided at the begining of the text or explained when they are used in Tables or Figures.  

It would be interesting that some images of the samples are provided. 

Author Response

1) Please, change the title of the paper. The term RCA used is not explained in the title. The readers cannot know the meaning.

As pointed out, “RCA concretes” was changed to “Recycled Aggregate Concretes.”

2) In the same way, add the explanation of acronyms in the abstract.

As suggested, “RCA” was corrected to “RCA (recycled aggregate concretes)”, in the Abstract.

3) The main question is why this cement paste dissociation agent used? Can the authors explain the composition? Preliminary studies should be more detailed to know how the authors consider that formulation.

As requested, the mechanism behind the increase of concrete due to the incorporation of CPDA was further described in Section 2.3 Cement Paste Dissociation Agent as follows:

“Free calcium oxide of cement forms calcium hydroxide when mixed with water. Then calcium hydroxide takes part in reactions with the second and forth components to the left in the following equations which are main component of the CPDA further to form inorganic crystallization or ettringite around old ITZs or new ITZs.”

And the followings were added to explain the background how our authors selected the 2.5% and 5% CPDA dosages as a percentage of cement amounts:

“1.25 % of CPDA powder as a percentage of cement amounts was recommended for conventional cement concretes by the CPDA manufactured company [46].  However, in this study to coat the weak parts of old ITZs in the RCA concrete, 2.5% of CPDA powder as a cement percentage, which is two times higher than the recommended dosages, was selected. Thus, for a preliminary test program (first mix series in Table 2), 0, 2.5, and 5% of CPDA powder as a percentage of cement amounts were added in RCA concrete mixes.”

4) The nomenclature used by the authors is difficult to understand. A list of acronyms should be provided at the beginning of the text or explained when they are used in Tables or Figures. 

As suggested, the nomenclature was explained with parenthesis right after the abbreviated terms.

5) It would be interesting that some images of the samples are provided.

As also suggested by Reviewer 3, a new section 4.2.3. Micro structural analysis of RCA was added and image analysis through optical microscope was described.

Reviewer 2 Report

Comments This paper investigated the drying shrinkage and rapid chloride penetration resistance of RCA concretes using cement paste dissociation agent. The outcome is interesting for readers. However, there are several aspects that need to be improved. The reviewer can only recommend for publication if the author satisfactorily address the following comments in the revised version. 1. How much improvement of mechanical properties was observed with the addition of 2.5% of CPDA? This need to be included in conclusion section. 2. Conclusion 2 and 3 may be merged together. 3. The mechanism behind the increase of strength of concrete due to the incorporation of CPDA need to be explained. 4. The novelty of the study should be highlighted in the end of introduction section. How this study is different from the published study in literature? 5. How the outcome of this study will benefit researchers and end users? This need to be highlighted in introduction or end of conclusion. 6. The background study on the strength development process of concrete containing waste materials is insufficient. The growing interest of waste materials in concrete construction need to be highlighted. The recent investigation on the strength development process in normal concrete [Ref: Characteristics, strength development and microstructure of cement mortar containing oil-contaminated sand] and 3D-printed concrete [Ref: 3D-printed concrete: applications, performance, and challenges] are particularly important. Suggest to include them in introduction section with proper citations to improve the background study. I would be happy to see the revised version to understand how these comments are being addressed.

Author Response

1) How much improvement of mechanical properties was observed with the addition of 2.5% of CPDA? This need to be included in conclusion section.

As pointed out, improvement of mechanical properties with the addition of 2.5% of CPDA were described as percentage increase, compared to the control samples as follows:

(1)… “Test results showed that about 24%, 18%, and 3% increase were observed for compressive strength, flexural strength, and elastic modulus, respectively at 28 days.”

2) Conclusion 2 and 3 may be merged together.

As suggested, Conclusion (2) and (3) were merged as follows:

“(2) All RCA concretes mixed with coated RCA were found to be workable regardless of different mix methods, with their slump and air contents of all the mixes being almost identical. And all the concrete specimens, which were mixed with the coated RCAs with CPDA solution, represented lower drying shrinkage values and RCPT (Rapid Chloride Penetration Test) values than those mixed without RCA coating treatment, regardless of different mix proportioning methods or RCA replacement ratios. And the variation of compressive strength and density of the RCA concrete did not affect drying shrinkage nor RCPT values.”

3) The mechanism behind the increase of strength of concrete due to the incorporation of CPDA need to be explained. 

As requested, the mechanism behind the increase of concrete due to the incorporation of CPDA was further described in Section 2.3 Cement Paste Dissociation Agent as follows:

“Free calcium oxide of cement forms calcium hydroxide when mixed with water. Then calcium hydroxide takes part in reactions with the second and forth components to the left in the following equations which are main component of the CPDA further to form inorganic crystallization or ettringite around old ITZs or new ITZs.”

4) The novelty of the study should be highlighted in the end of introduction section. How this study is different from the published study in literature?

As pointed out, the novelty of this study was highlighted at the end of 1. Introduction as follows:

“The above studies highlighted especially the importance of idealized mixing process and mix proportioning method, ITZ strengthening and coating method to provide the enhanced mechanical properties for RCA concretes. However, most of previous studies has been limited to investigate the mechanical strength properties of RCA concretes. Readers may notice that so far, no research to determine the effect of RCA coating treatment by cement type solutions has been conducted on durability properties such as drying shrinkage and chloride ion penetration resistance. Thus, in the present study, an RCA coating treatment by a cement paste dissociation agent (CPDA) with different mixing methods was newly incorporated in RCA concrete mixtures.”

5) How the outcome of this study will benefit researchers and end users? This need to be highlighted in introduction or end of conclusion.

As pointed out, the beneficial outcome of this study was described at the end of 1. Introduction as follows:

“Therefore, the results of this study will provide guidance that can be used to assess the beneficial increment of durability properties by adoption of RCA coating treatment with the optimized mix proportioning method.”

Reviewer 3 Report

Please see the file attached.

Author Response

1) Introduction focuses only on the RCA application to concrete. In this part of the article, the authors may add the information that RCA is only one of many types of the recycling aggregates that are used for concrete. A brief mention should be made of other recycling aggregates, e.g. glass waste, ceramic waste, rubber waste, wood waste, etc. Examples of references to the literature: 

As suggested, the followings were added in the Introduction section and several references were added.

“It is widely acknowledged that various types of waste materials can be transformed into recycled aggregates, powders, or additives and being used in concrete. Ceramic materials have been used as ceramic powder and ceramic aggregate in concrete [1]. The issue of using glass wastes into the concrete production and its advantages were summarized well by ZegardÅ‚o et al.[2]. And the recent studies related to the use of rubber aggregates and chips were reported by Guettla et al.[3]. Wood chips can be used for wood plastic composites paver blocks [4].”

Rahid, K.; Razzaq, A.; Dhmad, M.; Rashid T.; Tariq S. Experimental and analytical selection of sustainable recycled concrete with ceramic waste aggregate. Constr. Build. Mater, 2017, 154, 829-840.

Szelag, M; Zegardło, T.; Andrzejuk W. The use of fragmented, worn-out car side windows as an aggregate for cementitious composites. Materials, 2019, 12(9), 1467.

Ramdani S.; Guettala A.; Benmalek M.; Agular J. Physical and mechanical performance of concrete made with waste rubber aggregate, glass powder and silica sand powder. J. Build. Eng., 2019, 21, 302- 311.

Yang, S. A feasibility study of wood-plastic composite paver block for basic rest areas, J.  Kor. Wood Sci. Technol. 2019, 47(1), 51-65.

2) The aggregate particle size distribution is of key importance for the properties of concrete, because a properly composed crumb pile indicates the integrity of the mixture. Please provide the particle size distribution curves of all aggregates used in the tests.

As suggested, a new “Fig. 2. Aggregate gradation for RP, NCA, and fine aggregate 2” was added and a short explanation was given in the text as follows:

“Figure 2 shows the particle size distribution for the RP, NCA, and fine aggregate 2, satisfying Korean standards.”

3) Fig. 5, Table 3 – Please provide information regarding the dispersion of the results, i.e., standard deviation or coefficient of variation. Without this data, it is not possible to assess the reliability and repeatability of the obtained results.

As suggested, in Fig. 5, error bar was provided and explained in the main text. And in Table 3, coefficient of variation was added for the concrete compressive strengths.

4) The article has the characteristics of a technical paper rather than scientific nature. The authors limit their analyzes to comparing the results for individual recipes. The study lacks a detailed scientific discussion of the results and an attempt to explain why the results were obtained and not others. There is no link between the results of mechanical properties and the transformation of the microstructure of the tested concretes. The research should be supplemented with chemical and compositional analyzes, e.g. XRF, XRD, SEM-EDS, thermogravimetry, etc.

As also suggested by Reviewer 1, a new section 4.2.3. Micro structural analysis of RCA was added and image analysis through optical microscope was described.

Round 2

Reviewer 2 Report

I have no further comments.

Author Response

s the author of the paper, I fully appreciate all the feedback that you have given. It would improve the quality of the paper greatly.

Reviewer 3 Report

Almost all comments were addressed and included in the revised version of the manuscript. Photos shown on Fig. 16 are of very bad quality. It seems to me that they add little to the article.

Author Response

As the author of the paper, I fully appreciate all the feedback that you have given. It would improve the quality of the paper greatly. In order to solve the problems you have pointed out, which was to apply image process, we sought out for many institutions. It took a considerable amount of time to do so, and yet another much time was spent for the MDPI publisher to revise, combined with the two weeks time of grace period. To summarize, there was not enough time for image process, and the quality of the image could have been greatly improved if there were more time. The publisher only gave me two days for turning in the paper. I worked on improving the clarity of the image. I will try to improve the clarity of the image as much as possible during proofreading process. Once again, I am grateful to you for such detailed feedback.